# Cell Lysis Directed by SulA in Response to DNA Damage in *Escherichia coli*

**DOI:** 10.3390/ijms22094535

**Published:** 2021-04-26

**Authors:** Masayuki Murata, Keiko Nakamura, Tomoyuki Kosaka, Natsuko Ota, Ayumi Osawa, Ryunosuke Muro, Kazuya Fujiyama, Taku Oshima, Hirotada Mori, Barry L. Wanner, Mamoru Yamada

**Affiliations:** 1Life Science, Graduate School of Science and Technology for Innovation, Yamaguchi University, Ube 755-8611, Japan; muratam@yamaguchi-u.ac.jp (M.M.); tkosaka@yamaguchi-u.ac.jp (T.K.); b007vh@yamaguchi-u.ac.jp (N.O.); jyunroma@gmail.com (A.O.); 2Applied Molecular Bioscience, Graduate School of Medicine, Yamaguchi University, Ube 755-8505, Japan; keiko_nakamura038@yahoo.co.jp (K.N.); muro-im@m.u-tokyo.ac.jp (R.M.); dukes_blaze_soul_ignite@yahoo.co.jp (K.F.); 3Research Center for Thermotolerant Microbial Resources, Yamaguchi University, Yamaguchi 753-8515, Japan; 4Department of Biotechnology, Toyama Prefectural University, 5180 Kurokawa, Imizu, Toyama 939-0398, Japan; taku@pu-toyama.ac.jp; 5Graduate School of Information Science, Nara Institute of Science and Technology, 8916-5 Takayama, Ikoma, Nara 630-0192, Japan; hirotada.mori@gmail.com; 6Department of Microbiology, Harvard Medical School, Boston, MA 02115, USA; Barry_Wanner@hms.harvard.edu

**Keywords:** DNA damage, SulA, cell lysis, SoxS, LpxC

## Abstract

The SOS response is induced upon DNA damage and the inhibition of Z ring formation by the product of the *sulA* gene, which is one of the LexA-regulated genes, allows time for repair of damaged DNA. On the other hand, severely DNA-damaged cells are eliminated from cell populations. Overexpression of *sulA* leads to cell lysis, suggesting SulA eliminates cells with unrepaired damaged DNA. Transcriptome analysis revealed that overexpression of *sulA* leads to up-regulation of numerous genes, including *soxS.* Deletion of *soxS* markedly reduced the extent of cell lysis by *sulA* overexpression and *soxS* overexpression alone led to cell lysis. Further experiments on the SoxS regulon suggested that LpxC is a main player downstream from SoxS. These findings suggested the SulA-dependent cell lysis (SDCL) cascade as follows: SulA→SoxS→LpxC. Other tests showed that the SDCL cascade pathway does not overlap with the apoptosis-like and *mazEF* cell death pathways.

## 1. Introduction

DNA damage is harmful to all living organisms. Severe DNA damage leads to the removal of DNA-damaged cells from the cell population. In mammals, DNA damage directs the accumulation of the transcription factor p53, leading to cell cycle arrest or apoptosis. The p53 is stabilized by phosphorylation of its N-terminal in response to DNA damage [1,2,3], promoting transcription of the gene for the cyclin-dependent kinase inhibitor p21/ WAF1 [4], which inhibits the cyclin CDK2 complex, resulting in cell cycle arrest at G1 [5]. Severe DNA damage leads to a greater degree of p53 phosphorylation [6], leading to up-regulation of proapoptotic genes, such as p53AIP1 [7], which in turn leads to apoptosis.

DNA damage promotes an elaborate SOS response in *Escherichia coli* [8]. More than 40 genes are negatively controlled by LexA and up-regulated in response to DNA damaging agents [9], which in turn act to repair the damaged DNA [10] or cell death by LexA-controlled toxin-antitoxin genes [11]. The LexA regulon protein SulA inhibits assembly of FtsZ, an essential factor for cell division [12], arresting cell division and filamentous cell formation. Cell division arrest may supply time for DNA repair by DNA polymerases II, IV, V and other repair proteins [13]. Therefore, SulA may function as a checkpoint protein by inhibiting cell division when cells are exposed to DNA-damaging agents [14]. After completion of DNA repair, SulA is degraded by ATP-dependent proteases Lon and HslUV [15,16] and cell division resumes. However, serious defects in DNA’s over-repairing capability may lead to the removal of damaged cells by cell death/lysis.

The function of SulA in cell cycle arrest in *E. coli* is like that of p53 in eukaryotes, but it has not been clearly shown whether SulA or other factors are responsible for the elimination of highly damaged cells. We hypothesized that prolonged accumulation of SulA due to an unrepairable amount of damaged DNA causes cell death/lysis. Consistent with our hypothesis, overproduction of SulA resulted in protein accumulation in the medium, which was close to that of the treatment by a mutagen, mitomycin C (MMC) that is known to evoke an SOS response by introducing DNA damage—dG adduct of MMC or interstrand cross-linkage in DNA [17,18]—suggesting SulA-dependent cell lysis (SDCL). Further investigation including transcriptome analysis was conducted to find downstream factors in the SulA-dependent cell lysis (SDCL) cascade.

## 2. Results

### 2.1. Induction of Cell Lysis by DNA Damage

Because unrepaired damaged DNA is harmful not only for individual cells but also for the cell population, cell death/lysis may result when genomic DNA is severely damaged. To assess this in *E. coli*, changes of turbidity and colony-forming units (CFU) were examined in the *lon* mutant 5 h after the addition of MMC (Figure 1a). We assumed that SulA was a key factor to check DNA repair by preventing DNA replication until DNA damage had been cured. Compared to controls, the addition of MMC caused a sharp decrease in CFU with a 10,000-fold difference at 17 h (12 h after the addition of MMC) and a decrease in OD_600_ for 11 to 71 h (6 h to 66 h after the addition of MMC). Since the reduction in OD_600_ and flouting materials like those seen in phage lambda-directed lysis were like those of the σE-dependent cell lysis, which was seen in the early stationary phase and was enhanced by overexpression of the σE gene [19,20]; protein accumulation in the medium fraction after removal of the cell fraction by a low-speed centrifugation was analyzed by SDS-PAGE (Figure 1c). Strong protein bands were seen in samples from 17 h to 71 h (12 h to 66 h after the addition of MMC), but hardly seen in the control without MMC. In addition, when the same experiments with MMC were performed without supplementation of kanamycin, similar intensity patterns of protein bands were observed, indicating that observed effects were not influenced by the cells being stressed by the antibiotic being present in the medium. These data suggest that cell lysis occurs when DNA is damaged.

### 2.2. Cell Lysis Triggered by Overexpression of sulA

Effects of *sulA* overexpression on cell growth and cell lysis were examined. To avoid degradation of SulA by Lon protease, experiments were done with a *lon* mutant harboring pBAD-*sulA*, in which *sulA* was expressed from the *araBAD* promoter. The *sulA* overexpression induced by L-arabinose led to a reduction in OD_600_ and a 100-fold decrease in CFU at 14 h, corresponding to 9 h after arabinose addition (Figure 1b) and protein accumulation in the medium fraction (Figure 1d). Reductions in OD_600_ and CFU were like those by MMC, but the latter gave stronger effects. To confirm cell lysis, β-galactosidase activity was measured in the medium fraction as a cytoplasmic enzyme (Appendix A). Higher β-galactosidase activities were seen with *sulA* overexpression. Together these results imply that SulA accumulation causes cell lysis.

### 2.3. Morphological Observation and Live/Dead Staining of Cells Grown under the Condition with MMC or of Overexpression of sulA

Morphological changes by MMC and *sulA* overexpression occurred due to the SulA inhibition of cell division (Figure 2). Filamentous cells were seen at 2 h and 24 h after arabinose addition for *sulA* overexpression and at 2 h after MMC treatment, after which the numbers decreased drastically and rod-shaped cells were dominant at 24 h. RT-PCR analysis showed *sulA* expression level by MMC was about 32-fold lower than by arabinose at 3 h (Appendix A). Therefore, the lower expression of *sulA* but similar or greater protein accumulation by MMC (Figure 1) suggests that there exists both SulA-dependent (SDCL) and SulA-independent (SICL) cell lysis pathways induced by DNA damage. Notably, W3110N Δ*lon sulA*::*kan* cells treated by MMC showed protein accumulation at a level like that of the wild type treated with MMC (Appendix A), but CFU did not recover to the control levels even after 48 h unlike W3110N *lon*::*kan* (Figure 1). The former results allow us to speculate that the SICL pathway is enhanced in the *∆sulA* background and the latter suggests that *sulA* is required for recovery after MMC treatment.

### 2.4. Exploration of SulA Downstream Genes in the SDCL Pathway

To uncover SulA downstream genes in the SDCL pathway, total RNA was isolated at 3 h after induction of *sulA* expression by arabinose and subjected to DNA microarray analysis. The *sulA* expression level was about 150-fold higher than in the no-arabinose control (Appendix A). A total of 62 genes were up-regulated >2-fold and 51 were down-regulated, i.e., <0.5-fold (Appendix A). Up-regulated genes included 11 in energy metabolism and 18 for translation; down-regulated genes included 8 for central intermediary metabolism and 4 for energy metabolism. Fewer genes belonged to other cellular processes (Appendix A).

Among candidates for SulA downstream genes (Appendix A), four that were significantly upregulated were chosen for further analysis (*soxS*, *sodA*, *helD* and *cspB*). Of these, *soxS*, *sodA* and *helD* are related to responses for oxidative stress or DNA damage. Additionally, *cspB* was selected because it was the most highly up-regulated gene among five belonging to the cold shock response family: *cspB*, *cspF*, *cspG*, *cspH* and *cspI* (Appendix A), and like *cspA* [21] may respond to other stresses.

### 2.5. Effects of Deletion and Overexpression of soxS, sodA, helD and cspB

To examine whether *soxS*, *sodA*, *helD* and *cspB* lie downstream from SulA in the SDCL pathway, BW25113 mutants deleted of these genes (Appendix A) carrying pBAD-*sulA* were examined. As BW25113 can take up but not catabolize L-arabinose due to the Δ*araBAD* mutation (Appendix A), induction of *sulA* expression by L-arabinose was expected to continue for longer periods. As a result, the Δ*helD*, Δ*cspB* and Δ*soxS* mutants, but not the Δ*sodA* mutant, showed significantly lower protein accumulation compared to BW25113 (Figure 3a). However, Δ*helD*, Δ*cspB* and Δ*soxS* mutants showed higher turbidities than those of the wild type and Δ*sodA*, thus their low levels of protein accumulation were not due to their growth defects. Further experiments with BW25113 harboring pBAD-*helD*, pBAD-*cspB* or pBAD-*soxS* (Figure 3b) revealed protein accumulation in the medium fraction for cultures in which *helD* or *soxS* was overexpressed. These results suggest that HelD and SoxS lie downstream from SulA in the SDCL pathway and contribute the most. Considering these findings, no further experiments on other significantly up-regulated and down-regulated genes were performed.

### 2.6. Induction of soxS by sulA Overexpression

An earlier report showed that *soxS* induction occurs in response to an increase in reactive oxygen species (ROS) [22], in which SoxR up-regulates *soxS* in response to redox-active compounds under oxidative stress conditions [23]. Therefore, the intercellular ROS level was checked with the fluorescence probe H_2_DCFDA after *sulA* overexpression. Results showed that upon induction with L-arabinose, ROS levels were similar in BW25113 cells harboring pBAD-*sulA* and in BW25113 control cells harboring an empty vector at 0 h, 1 h, 3 h, 6 h and 12 h after induction. Yet, the Δ*soxR* mutant harboring pBAD-*sulA* showed similar protein band intensities in the medium fraction as the parental control harboring pBAD-*sulA* (Appendix A), suggesting that SoxR is not involved in the SDCL pathway. This finding is not unexpected because the SoxS regulon consists of genes under SoxS control and ones under the control of both SoxS and SoxR [24].

### 2.7. Exploration of SoxS Downstream Genes in the SDCL Pathway

As a DNA-binding dual transcription regulator, SoxS activates expression of 19 genes involved in defense against oxidative stress [24]. Thus, 16 non-essential SoxS downstream genes were examined with the corresponding deletion mutants harboring pBAD-*sulA*. Samples were taken from cultures at 24 h after L-arabinose induction where all strains except for Δ*helD* and Δ*soxS* mutants harboring pBAD-*sulA* showed almost similar levels of turbidity, and medium fractions were subjected to SDS-PAGE analysis. Results showed these mutants had only slight reductions of protein band intensities when compared with the control and greater intensities when compared with Δ*helD* and Δ*soxS* mutants harboring pBAD-*sulA*, as added controls (Appendix A). These data show that none of the 16 non-essential SoxS downstream genes are related to SDCL. Other experiments were then conducted to check for the involvement of the three essential genes of the SoxS regulon (*ribA*, *lpxC* and *fldA*) in the SDCL pathway.

### 2.8. Overexpression of Essential Gene Candidate of the SoxS Regulon

BW25113 harboring pCA24N-*fldA*, pCA24N-*ribA* or pCA24N-*lpxC* were examined by cell lysis assay after inducing gene expression with isopropyl β-D-thiogalactopyranoside (IPTG). Turbidities of BW25113 harboring pCA24N-*fldA* or pCA24N-*ribA* at 24 h were higher than that of BW25113 harboring pCA24N-*lpxC*. Results showed that cells carrying pCA24N-*lpxC* showed stronger protein intensity bands than those carrying pCA24N-*fldA* or pCA24N-*ribA* (Figure 4a). However, a large amount of a 28 kDa band was seen for pCA24N plasmids. To eliminate the possibility that the 28 kDa protein is responsible for cell lysis, BW25113 harboring pBAD-*lpxC* was evaluated. Overexpression of *lpxC* showed much stronger protein band intensities for the medium fraction than control cells. Moreover, medium protein band intensities were comparable for BW25113 harboring pBAD-*sulA*, pBAD-*soxS*, and pBAD-*lpxC* (Figure 4b). Greater protein band intensities for BW25113 carrying pBAD-*lpxC* may result from difference in expression of *lpxC*, due to the distance of LpxC from the trigger far downstream from SulA. Taken together, these results suggest that LpxC is a SoxS downstream factor in the SDCL pathway.

### 2.9. Effect of an Inhibitor for LpxC on SDCL

Since no Δ*lpxC* mutant exists because *lpxC* is an essential gene, effects of the LpxC-specific inhibitor CHIR-090 (MedChemExpress) were checked in cells overexpressing *sulA* or *soxS* (Figure 5). When cells harboring an empty vector were grown in the presence of 100 ng/mL or 1000 ng/mL of CHIR-090, strong medium fraction protein bands were seen, suggesting cell lysis by CHIR-090. Lysis may be due to severe inhibition of LpxC to a level below its requirement for cell proliferation. Further, CHIR-090 treatment of cells overexpressing *sulA* or *soxS* reduced protein band intensities in the medium fraction, implying that induction of LpxC synthesis by *sulA* or *soxS* overexpression was inhibited thereby reducing LpxC-directed cell lysis, but not cell proliferation due to some LpxC molecules escaping the inhibitor. These findings together with the results from overexpression of *lpxC* suggest that LpxC lies downstream from SoxS and takes part in the SDCL pathway.

### 2.10. Relationship of the SDCL Pathway with Known Cell Death Pathways

There are two known cell death pathways triggered by DNA damage in *E. coli*: *mazEF*-mediated cell death pathway [25] and the apoptosis-like death (ALD) pathway [26]. The former blocks operation of the latter. The latter is characterized by membrane depolarization and DNA fragmentation, which are hallmarks of eukaryote mitochondrial apoptosis and up-regulation of a unique set of genes called extensive-damage-induced (*Edin*) genes [26].

Studies focused on *mazEF* and *rybB* as key genes, which are analogous to the *Edin* genes because *rybB* is a key gene in the σE-dependent cell lysis pathway, which is triggered by ROS in the early stationary phase [20,27]. Experiments with BW25113Δ*mazEF* harboring pBAD-*sulA* and BW25113Δ*rybB* harboring pBAD-*sulA* were thus performed. Strong medium protein band intensities like those of BW25113 harboring pBAD-*sulA* as a control were from both strains (Appendix A). These results and the finding that ALD is tightly associated with the formation of OH˙ [26] but almost no increase in ROS occurs in SDCL suggest that the SDCL pathway does not overlap with *mazEF*-mediated cell death pathway nor with the ALD pathway.

## 3. Discussion

Based on the idea that severely DNA-damaged cells are eliminated from cell populations, we focused on SulA as the trigger that acts as a checkpoint protein to block cell division by inhibition of FtsZ polymerization in response to DNA damage in *E. coli* [12,14] and investigated the effects of *sulA* overexpression and SulA accumulation on cell morphology, cell death and lysis. Analysis of protein accumulation in the medium fraction revealed that overexpression of *sulA* resulted in cell lysis to the extent of that found by MMC treatment (Figure 1). The lower expression of *sulA* but similar or greater protein accumulation by MMC and protein accumulation in the Δ*sulA* background imply that there are also SICL pathways. Our findings suggest that SICL pathways include the ALD [26] and *mazEF*-cell death pathways [25] because both are induced by DNA damage and shown not to overlap with SDCL (Appendix A).

Factor(s) downstream from SulA in the SDCL pathway were investigated by transcriptome analysis to find gene candidates downstream of SulA. Gene candidates were examined by using the respective deletion mutants and by overexpression, suggesting SoxS together with HelD are found downstream from SulA. However, no increase in ROS was found with *sulA* overexpression; nor was an involvement of SoxR in SDCL (Appendix A). Further experiments with 19 deletion and overexpression strains for genes in the SoxS regulon suggested that LpxC is a main player downstream from SoxS in SDCL. Cell lysis caused by *helD* (encoding helicase IV that interacts with ssDNA) overexpression is consistent with the earlier report that even a modest level of *helD* overexpression may be lethal [28]. Additionally, *helD* overexpression causes a filamentous cell morphology [29] and may enhance the filamentous phenotype by SulA. There is no further evidence on the factors downstream from HelD in HelD-dependent cell lysis.

Earlier reports gave a line of evidence to explain the cell lysis by overproduction of LpxC. Sutterlina et al. [30], reported a novel cell death pathway triggered by a dominant *mlaA* mutation that removes phospholipids from the outer leaflet of the outer membrane (OM), in which LpxC may be involved. LpxC catalyzes the first committed step in LPS biosynthesis and the level of LPS is tightly regulated by FtsH/LapB/LapC complex and HslUV protease [31,32] and excessive production of LPS is toxic to cells [33,34]. The *ftsH1* (Ts) mutation increases the amount of lipopolysaccharide at the non-permissive temperature due to a dramatic increase for LpxC, UDP-3-O-(R-hydroxymyristoyl)-N-acetylglucosamine deacetylase [33]. It has been proposed that the *mlaA* allele-directed cell death is initiated by destabilization of the OM to produce OM vesicles and subsequently, the lipids lost from the OM are replaced by the lipids from the inner membrane (IM), resulting in a shrink of the IM, reduction of the cytoplasmic volume and increase in its density, which finally, leads to the mechanical rupture of the IM and cell lysis [30]. As in this proposal, the increased level of LPS by overproduction of LpxC may lead to destabilization of the OM and initiation of cell lysis.

Based on our findings, we propose the SDCL pathway model for cell death and lysis by DNA damage that is shown schematically in Figure 6. Our study suggests that the SDCL pathway does not overlap with the *mazEF*-mediated [25] or ALD pathway [26], which are also induced by DNA damage. In the SDCL pathway, SoxS activates *lpxC* expression without any factor(s) linking SulA and SoxS. Measurements of ROS revealed that overexpression of *sulA* caused a slight change in ROS levels, while *sodA* was induced under the same conditions. These inconsistent findings are understandable if a trace amount of ROS can induce *sodA* or the negative regulation of *sodA* by Fur or ArcA [35] is released. In addition, it is known that the induction of *soxS* occurs not only by oxidative stress but also via cross-regulation by other transcription regulator(s) [36]. DNA damage induces *sulA* expression as an SOS response. If SulA exists stably due to DNA damage being too severe to be completely repaired, the SDCL pathway, composed of SulA, SoxS, and LpxC, is evoked to promote cell lysis. Further research, however, is required to show a direct link between SulA, SoxS and LpxC. On the other hand, HelD is responsible for alternative cell lysis in the SDCL pathway, which is different from the SoxS-directed cell lysis.

## 4. Materials and Methods

### 4.1. Bacterial Strains, Medium and Culture Conditions

Bacterial strains used in this study were derivatives of *E. coli* K-12. Their relevant genotypes and plasmids are shown in Appendix A. For experiments with gene-disrupted mutants, BW25113 derivatives [37] or W3110N [38] derivatives were used. The transfer of *lon*::*kan* or *sulA*::*kan* from BW25113 *lon*::*kan* or BW25113 *sulA*::*kan* to W3110N, and the transfer of *sulA*::*kan* from BW25113 *sulA*::*kan* to W3110N Δ*lon* were performed by P1 transduction according to the procedure previously described [39]. The construction of *mazEF*::*kan* in BW25113 and the removal of *kan* from W3110N *lon*::*kan* and W3110N *sulA*::*kan* were performed with pCP20 according to the procedure previously described [40]. The first generated allele of *mazEF*::*kan* was further transferred into BW25113 by P1 transduction, generating BW25113 *mazEF*::*kan*. The construction was confirmed by PCR with the genomic DNA of the transductant as a template. For gene overexpression, pCA24N recombinants of specific genes under control of an IPTG-inducible promoter in an ASKA library [41] or pBAD24 [42] recombinants of specific genes under control of an L-arabinose-inducible promoter were used. Liquid culture was performed by using modified Luria–Bertani (LB) medium (1% Bacto Tryptone (Nacalai Tesque, Japan), 0.5% yeast extract (Nacalai Tesque, Japan), 0.5% NaCl) at 37 °C under aerobic conditions by reciprocal shaking (100 times/min). In growth experiments, precultured cells were inoculated into LB (0.1% of total volume), and cell growth was seen by monitoring turbidity or CFU. Appropriate antibiotics were added at the following final concentrations: ampicillin (Meiji Seika, Japan), 50 µg/mL; chloramphenicol (Wako, Japan), 20 µg/mL; and kanamycin (Meiji Seika Kaisha, Japan.), 25 µg/mL. MMC (Wako, Japan) was added as a mutagen at a final concentration of 0.1 µg/mL. L-arabinose and IPTG were added at final concentrations of 0.1% and 0.1 or 0.01 mM, respectively, into the culture at around OD_600_ of 0.5 to induce the expression of genes that had been cloned.

### 4.2. DNA Manipulation

Conventional recombinant DNA techniques [43] were applied for DNA manipulation. For construction of pBAD recombinants, DNA fragments bearing the coding regions of *sulA*, *soxS*, *lpxC* and *helD* were amplified by polymerase chain reaction (PCR) using primers with an *Eco*RI or *Hin*dIII recognition sequence and genomic DNA as a template. The primers used are shown in Appendix A. The amplified PCR products were purified by using a QIAquick PCR Purification kit (Qiagen, The Netherlands). The DNA fragments of pBAD24 after digestion with *Eco*RI and *Hin*dIII were also purified by using the same kit and were treated with Shrimp alkaline phosphatase (TAKARA, Japan) followed by heating to inactivate the enzyme. The fragments were further purified by using the same kit. Both fragments were ligated by using Ligation high Ver.2 (Toyobo, Japan). The ligated materials were introduced into Competent high DH5α (Toyobo, Japan). Similarly, for construction of pCAN24 recombinants, DNA fragments bearing the coding regions of *sulA* and *soxS* were amplified by PCR using one primer with a *Not*I recognition sequence and another primer without a restriction enzyme recognition sequence and genomic DNA as a template. The fragments were digested with *Not*I and ligated with *Not*I and *Stu*I double digested pCA24N fragments. Plasmid DNAs in transformants were purified by using a QIAprep Spin Miniprep kit (Qiagen, The Netherlands). The construction of plasmid recombinants was confirmed by restriction mapping.

### 4.3. Analysis of Proteins in the Medium Fraction

During cultivation, 1.0 mL of the culture medium was taken and subjected to centrifugation at 3000 rpm for 10 min to separate the supernatant (medium fraction) and the precipitate. Proteins in the medium fraction were recovered by centrifugation at 3000 rpm for 10 min after the addition of trichloroacetic acid at a final concentration of 10% and kept on ice for 30 min. The precipitate was washed with diethyl ether and resuspended with 100 µL of 20 mM Tris-HCl (pH 7.0). After addition of sodium dodecyl sulfate (SDS) at a final concentration of 1% and 2-mercaptoethanol at a final concentration of 0.5%, 30 µL of the medium fraction was heated at 70 °C for 10 min with a small amount of sucrose and then applied onto SDS-12% PAGE. The proteins separated on the gel were stained with Coomassie brilliant blue R250.

### 4.4. Reverse Transcriptase-Polymerase Chain Reaction (RT-PCR)

Total RNA was prepared from W3110N *lon*::*kan* cells grown without or with MMC or from W3110N *lon*::*kan* cells harboring pBAD-*sulA*, which were grown in LB medium for 1 h and 3 h after induction at 37 °C, by the hot phenol method [44]. The concentration of RNA was estimated spectrophotometrically at 260 nm. RT-PCR analysis was performed by using the mRNA Selective RT-PCR Kit (TAKARA, Japan) with 0.1 µg of RNA as a template and primer sets (Appendix A) to examine the expression of targeted genes as performed previously [45]. RT reaction was conducted at 45 °C for 15 min with 0.1 µg of total RNA and each downstream primer, and then, PCR reaction was conducted at 85 °C for 1 min for denaturing, at 45 °C for 1 min for annealing and at 72 °C for 1 min for extension by using the two primers for each gene. The PCR products after 15, 20, 25 and 30 cycles were analyzed by 0.9% agarose gel electrophoresis and stained with ethidium bromide. As a control, 10 µg of total RNA was run by 1.2% agarose gel electrophoresis, followed by staining with ethidium bromide. The intensity of bands of RT-PCR products was quantitatively determined using ImageJ. Linearity of the amplification was seen up to the 25th or 30th cycle. Under our experimental conditions, RT-PCR was able to specifically detect mRNA because no band was seen when reverse transcriptase was omitted.

### 4.5. Cell Staining and Morphological Observation

Cells were collected from 1 mL of the culture medium by centrifugation and washed with 50 mM potassium phosphate buffer (pH 6.5). After resuspension with the same buffer, staining of the cells was performed with Hoechist 33258 for observation of all cells, with Calcein-AM for the observation of viable cells and with propidium iodide (PI) (Wako, Japan) for observation of membrane-permeabilized cells (dead cells) for 15 min in the dark at concentrations of 5 µg/mL for Hoechist, 10 µg/mL for Calcein and 0.5 µg/mL for PI according to the supplier’s instructions. Samples were observed using a Nikon E600 microscope with fluorescence capability (Nikon, Japan).

### 4.6. Measurement of β-Galactosidase Activity

β-Galactosidase activity was measured according to the method of Miller [39]. During cultivation, 1.0 mL of the culture medium was taken and subjected to centrifugation at 3000 rpm for 10 min to separate the medium and cell fractions. Cells in the cell fraction were resuspended in LB medium and treated with toluene. The medium fraction and cell fraction were each mixed with Z buffer (16.1 g/L Na_2_HPO_4·_7H_2_O, 5.5 g/L NaH_2_PO_4_·H_2_O, 0.75 g/L KCl and 0.24 g/L MgSO_4_·7H_2_O (pH 7.0)). Enzyme reaction was started by the addition of *o*-nitrophenyl-β-D-galactopyranoside (Wako, Japan) and stopped by the addition of 1M Na_2_CO_3_. The intensities of color and turbidity were measured spectrophotometrically at OD_420_ and OD_550_ nm, respectively, and the activity of each sample was calculated as a Miller unit by using the established formula.

### 4.7. RNA Isolation and Preparation of DNA Microarrays

Total RNA was prepared from W3110N *lon*::*kan* cells harboring pBAD24 or pBAD-*sulA*, which were grown in LB medium containing ampicillin 3 h after induction by L-arabinose, by the hot phenol method [44]. After phenol/chloroform treatment and ethanol precipitation, the resultant RNA (about 100 μg) was resuspended in 100 mM sodium acetate, pH 5.5, 50 mM MgSO_4_ and treated at 37 °C for 1 h with 10 units of RNase-free DNAase (TAKARA, Japan) in a final volume of 100 µL. RNA was recovered after phenol/chloroform treatment and ethanol precipitation. The RNA concentration was estimated spectrophotometrically at 260 nm. DNA microarrays were prepared using synthetic 65-bp oligos designed for *E. coliI*-K12, O157 sakai and EDL genes (Sigma Genosys, Japan). The oligos were diluted with Pront universal microarray reagent (Corning, Glendale, AZ, USA) at 80 μM and spotted on ULTRA GAPS type 7 slides (Amersham Pharmacia Biotech, Inc., Erie, PA, USA) by using a Lucidea array spotter (Amersham Pharmacia Biotech, Inc., Erie, PA, USA). The spotted array DNAs were immediately UV-crosslinked with the array slides in UV stratalinker 1800 (Stratagene, La Jolla, CA, USA). Fluorescent-labeled cDNA preparation was performed with a Cyscribe post-labelling kit (Amersham BioSciences, UK) and a Cyscribe GFX purification kit (Amersham BioSciences, UK). Labelling of amino allyl-modified cDNA was performed with CyDYE (Amersham BioSciences, UK). The labeled cDNAs were purified with a Cyscribe GFX purification kit (Amersham BioSciences, UK). Before the hybridization of labeled probes with array DNAs, the DNA microarray slides were incubated in a prehybridization buffer (5 × SSC, 0.1% SDS, 1% BSA) at 42 °C for 45 min (prehybridization). The labeled probes were mixed with a probe solution (final concentration of 50% formamide, 6 × SSC, 0.6% SDS, 5 × Denhardt’s solution) and hybridized with array DNAs on DNA microarrays at 42 °C overnight. The slides were washed with Wash I (2 × SSC, 0.1% SDS), Wash II (0.2 × SSC) and Wash III solution at 42 °C for 2 min in each washing step. After the washing steps, the slides were dried and scanned with an FLA-8000 array scanner (Fiji Film, Japan) at 532 (Cy3) and 635 (Cy5) nm. Because our DNA microarray has spots including non-K-12 oligos (designed for pathogenic *E. coli*) and blank spots in a total of 12,288 spots, we selected 10 spots including the oligos for O157 genes ECs4964, ECs4965, ECs4966, ECs2142 and ECs2143 as negative control spots. To check the hybridization efficiency, the oligos for 16S rRNA were also spotted as positive control spots. The Cy3 and Cy5 fluorescent intensity values of each oligo were subtracted with the fluorescent intensity value calculated from negative control oligos (average + 1SD) in 10 negative control spots. The relative ratio of Cy5 and Cy3 was calculated and global normalization was performed for the average value of the log Cy5/Cy3 ratio of all oligos against *E. coli* K-12 genes to be 0. More than one oligos were designed for each gene and at least two independent spots were spotted for one oligo. Then, we selected the up or down oligos based on relative ratio of Cy5 and Cy3: lower (<0.5, i.e., a negative fold difference) or higher (>2.0, i.e., a positive fold difference) ratios of the *sulA* overexpression sample (Cy5) to the control sample (Cy3) were significant differences. Then we selected the genes corresponding to the up or down oligos as the up and down genes in the *sulA* overexpression sample. Functional classification of genes was performed according to the EcoCyc [46] and KEGG [47] databases. The functions of genes were derived or inferred from the GenoBase, SWISS-PROT and NCBI databases.

### 4.8. Measurement of Intracellular ROS

The amounts of ROS inside cells were measured with an oxidant-sensitive probe, 2′,7′-dichlorofluorescin diacetate, H_2_DCFDA (Sigma-Aldrich, St. Louis, MO, USA) [27]. This probe is trapped inside the cells after cleavage of the diacetate by an intracellular esterase. It is then oxidized by radical species (mostly H_2_O_2_) with a more fluorescent compound being produced. Cells were grown in LB medium containing H_2_DCFDA at a final concentration of 4 µM. Aliquots were then washed once with saline, resuspended in the same solution, and subjected to sonic oscillation. The cell extracts were centrifuged at 4000 rpm for 10 min and the supernatants were used. The fluorescence from the supernatants was measured with λ_EX_ = 502 nm and λ_EM_ = 524 nm using a fluorescence spectrophotometer (Hitachi, Japan). Protein concentration was determined by the Lowry method [48]. The value of λ_EM_ = 524 nm was normalized to the protein concentration in each sample.

### 4.9. Statistical Analyses

All experiments were performed independently at least three times, except for fluorescence microscopy and transcriptome analysis. The values of OD, CFU and β-galactosidase activity are shown with standard deviations. For morphology observation by a fluorescence microscope, we chose a typical image from at least ten random images. Transcriptome analysis and its statistical analysis were described above.

## Figures and Tables

**Figure 1 ijms-22-04535-f001:**
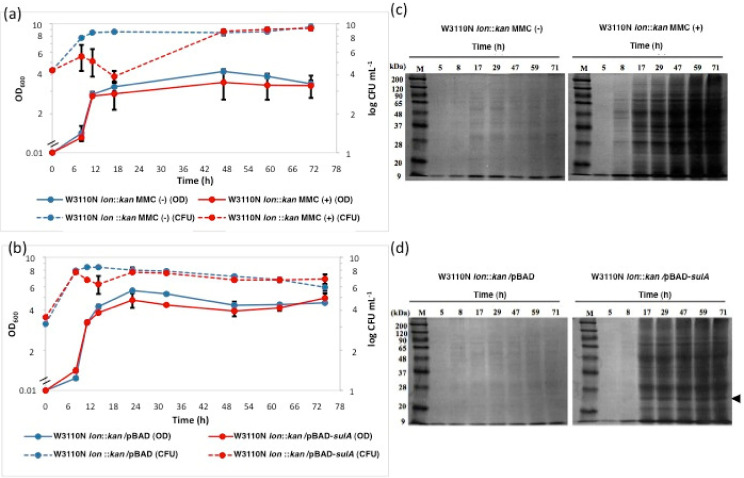
Effects of mitomycin C and overexpression of *sulA* on cell growth and cell lysis. W3110N *lon*::*kan* cells were grown in LB medium containing kanamycin (**a**) and W3110N *lon*::*kan* cells harboring pBAD24 or pBAD-*sulA* (**b**) were grown in LB medium containing ampicillin and kanamycin, and mitomycin C (MMC) and L-arabinose were added at 5 h (OD_600_ of about 0.5) at the final concentrations of 0.1 μg/mL and 0.1%, respectively. Cell turbidity (straight lines) and colony forming units (CFU) (dotted lines) were determined. Red circles and blue circles in (**a**) represent conditions with and without MMC, respectively; red circles and blue circles in (**b**) represent W3110N *lon*::*kan* cells harboring pBAD-*sulA* and W3110N *lon*::*kan* cells harboring pBAD24, respectively. Proteins from the medium fractions of cultures of W3110N *lon*::*kan* cells with and without MMC (**c**) and of W3110N *lon*::*kan* cells harboring pBAD-*sulA* and W3110N *lon*::*kan* cells harboring pBAD24 (**d**) were recovered and subjected to SDS-PAGE as described in the Materials and Methods Section. An arrowhead indicates the position of the size of SulA (18.8 kDa). Lane M is molecular markers. Error bars stand for ± SD.

**Figure 2 ijms-22-04535-f002:**
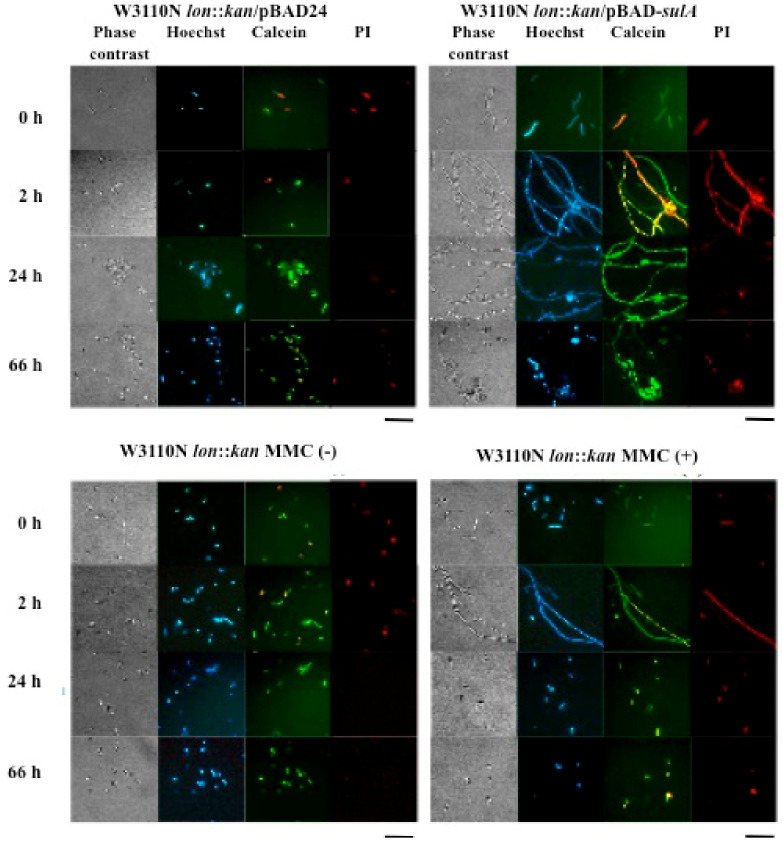
Morphological observation and live/dead staining of cells grown under the condition with MMC or with overexpression of *sulA*. W3110N *lon*::*kan* cells were grown in LB medium containing kanamycin; W3110N *lon*::*kan* cells harboring pBAD24 or pBAD-*sulA* were grown in LB medium containing ampicillin and kanamycin; MMC and L-arabinose were added at OD_600_ of about 0.5 at the final concentrations of 0.1 μg/mL and 0.1%, respectively. Microscopic observation of cells was performed directly at or after staining with Hoechst; Calcein-AM, or PI Bars are 10 μm.

**Figure 3 ijms-22-04535-f003:**
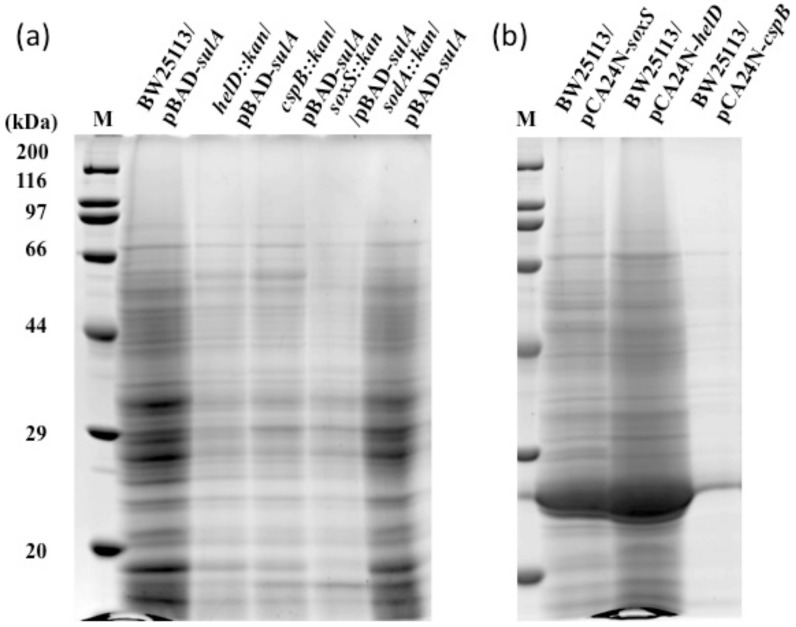
Effects of disruption or overexpression of selected up-regulated genes by overexpression by *sulA* on SDCL. BW25113 *helD*::*kan*, BW25113 *cspB*::*kan*, BW25113 *soxS*::*kan*, and BW25113 *sodA*::*kan* cells harboring pBAD-*sulA* (**a**), and BW25113 cells harboring pCA24N-*soxS*, pCA24N-*helD*, or pCA24N-*cspB* (**b**), were grown in LB medium with ampicillin. L-Arabinose or IPTG was added at OD_600_ of about 0.5 at the final concentrations of 0.1% and 0.1 mM, respectively. Cultures were sampled by removing portions at 24 h after induction. Sample preparation for SDS-PAGE was performed as described in the Materials and Methods Section.

**Figure 4 ijms-22-04535-f004:**
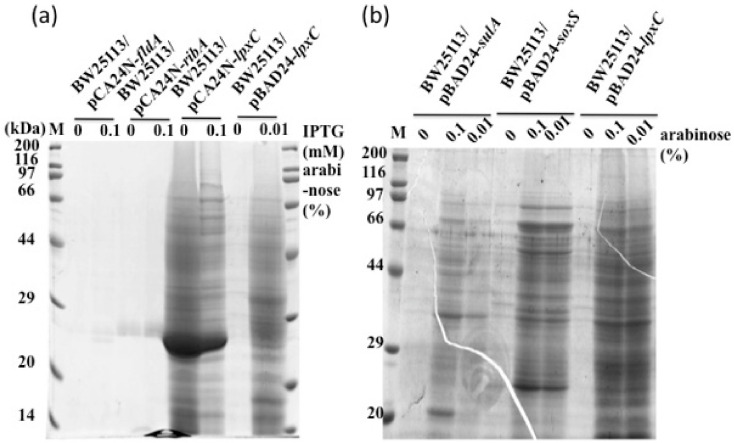
Effects of overexpression of selected SoxS regulon genes on SDCL. BW25113 cells harboring pCA24N-*fldA*, pCA24N-*ribA*, pCA24N-*lpxC* or pBAD-*lpxC* (**a**) and BW25113, BW25113 cells harboring pBAD-s*ulA*, pBAD-*soxS*, or pBAD-*lpxC* (**b**) were grown in LB medium containing ampicillin. L-Arabinose and IPTG were added at OD_600_ of about 0.5 at the final concentrations of 0.1% or 0.01% and 0.1 mM, respectively. Portions of cultures were taken at 24 h after induction. Sample preparation for SDS-PAGE was performed as described in the Materials and Methods Section.

**Figure 5 ijms-22-04535-f005:**
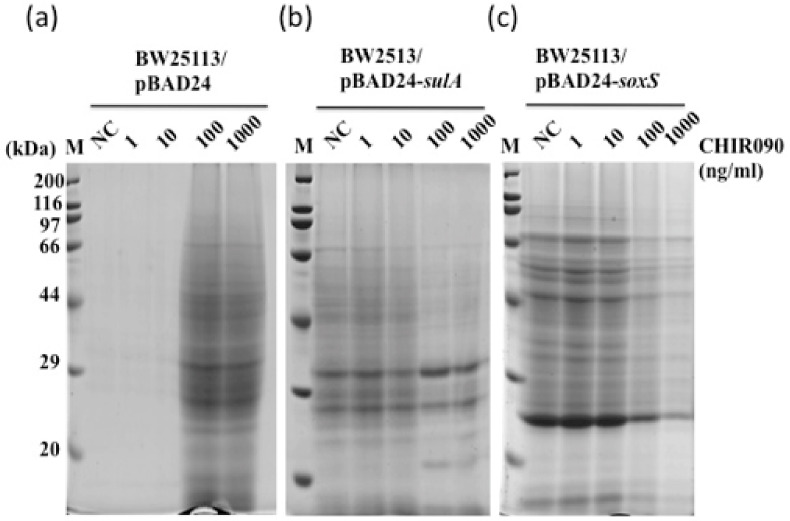
Effects of CHIR-090P, an LpxC inhibitor, on SDCL. BW25113 cells harboring pBAD24 (**a**), pBAD24-*sulA* (**b**) (SulA: 18.8 kDa) and pBAD24-*soxS* (**c**); (SoxS: 12.9 kDa) were grown in LB medium containing ampicillin. L-Arabinose (at a final concentration of 0.01%) and CHIR-090 (at a final concentration of 0, 1, 10, 100 or 1000 ng/mL) were added at OD_600_ of about 0.5, and portions of the culture were taken at 24 h after induction. Sample preparation for SDS-PAGE was performed as described in the Materials and Methods Section.

**Figure 6 ijms-22-04535-f006:**
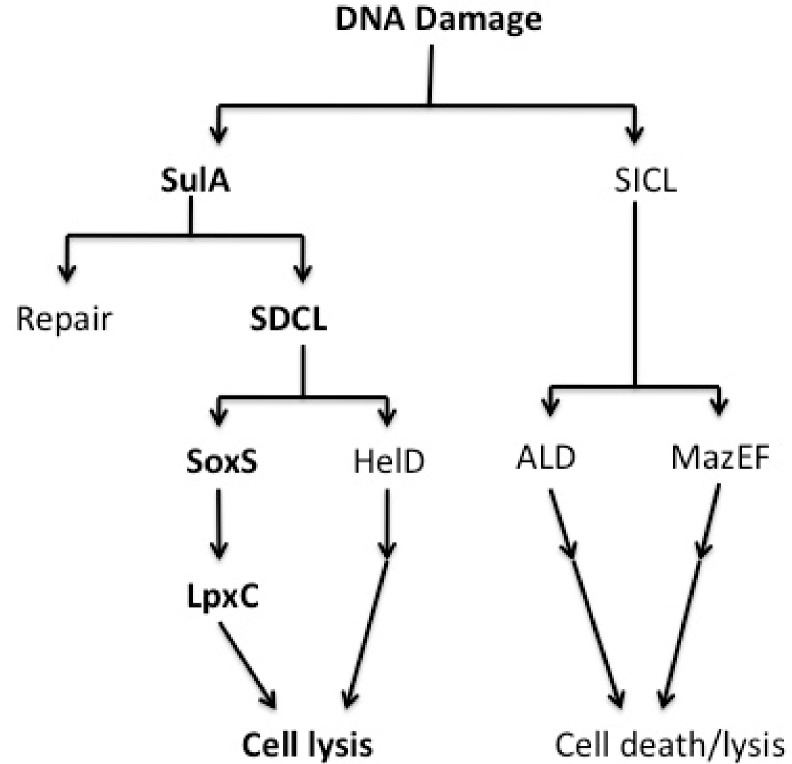
Pathways of SDCL following severe DNA damage. Details are described in the text.

## Data Availability

Array data are accessible through ArrayExpress accession number A-MTAB-684.

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
