# Peer review of "Cell Lysis Directed by SulA in Response to DNA Damage in *Escherichia coli"

_ijms, 2021, doi:10.3390/ijms22094535_

Round 1

Reviewer 1 Report

In this study Murata and co-workers describe experiments in E. coli that investigate how prolonged persistence of the division inhibitor SulA, which is expressed as part of the SOS response in bacteria, leads to cell lysis. The authors show the accumulation of proteins in the culture upon overexpression of the sulA gene, and effect also observed when cells are treated with MMC. SulA-overexpression results in persistent filamentation, while MMC treatment triggers filamentation for a period that corresponds with the removal of damage. The authors use microarray analysis to identify genes that show altered levels of transcription if SulA is over-expressed. It is then shown that in helD, cspB and soxS deletion mutants protein accumulation in the medium is reduced if SulA is over-expressed in these cells. The authors proceed to quite systematically analyse downstream effectors of these genes to narrow down the precise molecular effect for the lysis of cells. In the end they present quite a satisfying model of what might be the molecular reason in the Discussion.

Overall I thoroughly enjoyed studying the data presented. The authors do a very good job of putting the effect of damage-induced cell lysis into context, and the phenomenon has not received the attention it deserves.

However, in its current format the study is not yet sound, and additional work and clarification needs to be done before the work is ready for publication.

Major points

  1. Nomenclature needs to revisited throughout the manuscript. E. coli nomenclature is, unfortunately, not as clear as the eukaryotic nomenclature that clearly distinguishes between dominant and recessive alleles. Genes are consistently written lowercase and italics, something that is not adhered to, especially in the overexpression constructs (e.g. pBAD-SULA should be pBAD-sulA).
  2. Line 84, what is meant by “sE-dependent cells lysis”?
  3. Figure 1 panels a and b, these need a major clarification. It is an absolute nightmare to try and decipher what the graphs mean. The authors need to add clear descriptions to the panels, and the individual lines should receive labels in the graph, not just in the legend. Also, what is this figure trying to show? The similarities? Or the differences? Given that both MMC and SulA over-expression use “red and blue circles” it is impossible to get to the bottom of this image. Depending on what the aim is the datasets need to be presented as a direct comparison. If cells lyse – why is there no significant difference? The growth delay caused by filamentation is easily picked up by CFU, but why is cells lysis not detected?
  4. Figure 1, panel c (and most other figures), how do the authors normalise how much relative material is taken from the culture? It is only stated that 1 ml of the supernatant after centrifugation is taken. However, what if cell numbers differ significantly in the two cultures? This effect alone could result in higher protein levels. Obviously for some of the experiments CFUs have been done, but this not shown for other figures.
  5. Line 109, given the comparisons between SulA over-expression and MMC treatment, what happens if ∆sulA cells get MMC treated? What is the effect on lysis? Will simply the later described ALD pathway kick in?
  6. Figure 2, if SulA is over-expressed, why are filaments ever broken down? For MMC it is simply the removal of damage that is causing this effect, consistent with other studies. But what happens if SulA is continuously expressed? Also: can cell lysis be visualised via time-lapse microscopy?
  7. Figure 3, the normalisation is an issue here. Is there less material in the medium because there are significantly less viable cells? How can the authors compare between conditions?
  8. Figure 4, in the legend the authors refer to Figure 1, but in the legend for Figure 1 the authors refer to Material and Methods. The authors should find a better way here.
  9. Throughout the manuscript the authors state that for example ∆lon::kan cells were grown in medium with kanamycin. This is a very unfortunate misconception. Chromosomal deletions in E. coli are, with very few exceptions, fully stable and do not require any antibiotics for growth. Rather the opposite, including antibiotics routinely into the medium can, and does, cause unwanted side effects. Just as one example, if frozen kanamycin-resistant stocks are streaked on medium with kanamycin the cells are very significantly stressed and grow very poorly. I am not asking that the authors repeat all the experiments, but the authors need to demonstrate in an example experiment that the observed effects are not influenced by the cells being stressed by the antibiotic being present in the medium.
  10. The information in the strain table is insufficient. Simply stating “this study” does not tell the reader anything. The authors need to state the parent strain, the strain on which P1 was grown (for a transduction), and what was selected for. If new alleles are introduced (Datsenko and Wanner method), the authors must cross the newly-generated allele from the initial strain into a wild type background via P1 transduction, as “recombineering” procedures can result in secondary mutations. This also should be clearly reflected by the strain table.

Minor points

Line 58, E. coli need to be italicised.

Line 177, full stop missing after “sulA overexpression”.

Author Response

Reviewer I

Comments and Suggestions for Authors

In this study Murata and co-workers describe experiments in E. coli that investigate how prolonged persistence of the division inhibitor SulA, which is expressed as part of the SOS response in bacteria, leads to cell lysis. The authors show the accumulation of proteins in the culture upon overexpression of the sulA gene, and effect also observed when cells are treated with MMC. SulA-overexpression results in persistent filamentation, while MMC treatment triggers filamentation for a period that corresponds with the removal of damage. The authors use microarray analysis to identify genes that show altered levels of transcription if SulA is over-expressed. It is then shown that in helDcspB and soxS deletion mutants protein accumulation in the medium is reduced if SulA is over-expressed in these cells. The authors proceed to quite systematically analyze downstream effectors of these genes to narrow down the precise molecular effect for the lysis of cells. In the end they present quite a satisfying model of what might be the molecular reason in the Discussion.

Overall I thoroughly enjoyed studying the data presented. The authors do a very good job of putting the effect of damage-induced cell lysis into context, and the phenomenon has not received the attention it deserves.

However, in its current format the study is not yet sound, and additional work and clarification needs to be done before the work is ready for publication.

Major points (Corrected portions in the revised manuscript are shown in blue)

  1. Nomenclature needs to revisited throughout the manuscript. E. coli nomenclature is, unfortunately, not as clear as the eukaryotic nomenclature that clearly distinguishes between dominant and recessive alleles. Genes are consistently written lowercase and italics, something that is not adhered to, especially in the overexpression constructs (e.g. pBAD-SULA should be pBAD-sulA).

We have corrected as suggested.

  1. Line 84, what is meant by “sE-dependent cells lysis”?

The description has been improved as “the sE-dependent cell lysis, which was seen in early stationary phase and was enhanced by overexpression of the sE gene” in p 2.

  1. Figure 1 panels a and b, these need a major clarification. It is an absolute nightmare to try and decipher what the graphs mean. The authors need to add clear descriptions to the panels, and the individual lines should receive labels in the graph, not just in the legend. Also, what is this figure trying to show? The similarities? Or the differences? Given that both MMC and SulA over-expression use “red and blue circles” it is impossible to get to the bottom of this image.

According to the comments, the legend has been modified, and labels were added in the graph.

Depending on what the aim is the datasets need to be presented as a direct comparison. If cells lyse – why is there no significant difference? The growth delay caused by filamentation is easily picked up by CFU, but why is cells lysis not detected?

The values of OD are significantly different from each other.

To easily see the effect of cell lysis, the scale of vertical axis has been changed from logarithm to integer.

  1. Figure 1, panel c (and most other figures), how do the authors normalise how much relative material is taken from the culture? It is only stated that 1 ml of the supernatant after centrifugation is taken. However, what if cell numbers differ significantly in the two cultures? This effect alone could result in higher protein levels. Obviously for some of the experiments CFUs have been done, but this not shown for other figures.

As described in Materials and methods, 30% of samples prepared from the supernatant of 1 ml culture were subjected to SDS-PAGE. When compared, for example, between with MMC and without MMC in Figure 1c, cell number in the culture without MMC may be higher than that in the culture with MMC, but proteins in the culture without MMC were much lower that those in the case with MMC. Therefore, the effect of cell number alone may not result in higher protein level.

  1. Line 109, given the comparisons between SulA over-expression and MMC treatment, what happens if ∆sulA cells get MMC treated? What is the effect on lysis? Will simply the later described ALD pathway kick in?

W3110N Dlon sulA::kan cells treated by MMC showed protein accumulation at the level like that of the wild type by treated with MMC (Figure S3), but CFU did not recover to the control levels even after 48 h unlike W3110N Dlon::kan (Figure 1). The former results allow us to speculate that the SICL pathway is enhanced in the ∆sulAbackground and the latter suggests that sulA is required for recovery after MMC treatment. Related description to this has been added in p. 4.

New Figure S3 has been added.

  1. Figure 2, if SulA is over-expressed, why are filaments ever broken down? For MMC it is simply the removal of damage that is causing this effect, consistent with other studies. But what happens if SulA is continuously expressed? Also: can cell lysis be visualised via time-lapse microscopy?

Filamentous cells were dominantly detected even at 24 h where strong proteins bands from were detected at 17 h, suggesting that cell lysis occurs under the SulA over-expressed condition. Therefore, the continuous presence of SulA that indicates existence of unrepaired damaged DNA results in lysis by SDCL.

Unfortunately, we have no equipment for time-lapse experiments.

  1. Figure 3, the normalisation is an issue here. Is there less material in the medium because there are significantly less viable cells? How can the authors compare between conditions?

However, DhelD, DcspB and DsoxS mutants showed higher turbidities than those of wild type and DsodA and thustheir low levels of protein accumulation were not due to their growth defects. This description has been added in p 5.

Similarly, for Figure S4, a description of “Samples were taken from cultures at 24 h after L-arabinose induction where all strains except for DhelD and DsoxS mutants harboring pBAD-SULA showed almost similar levels of turbidity” has been added in p 6.

For Figure 4, a description of “Turbidities of BW25113 harboring pCA24N-FLDA or pCA24N-RIBA at 24 h after induction were higher than that of BW25113 harboring pCA24N-LPXC” has been added in p 6.

  1. Figure 4, in the legend the authors refer to Figure 1, but in the legend for Figure 1 the authors refer to Material and Methods. The authors should find a better way here.

According to comments, we have changed “Figure 1” to “Material and Methods”.

  1. Throughout the manuscript the authors state that for example ∆lon::kan cells were grown in medium with kanamycin. This is a very unfortunate misconception. Chromosomal deletions in E. coli are, with very few exceptions, fully stable and do not require any antibiotics for growth. Rather the opposite, including antibiotics routinely into the medium can, and does, cause unwanted side effects. Just as one example, if frozen kanamycin-resistant stocks are streaked on medium with kanamycin the cells are very significantly stressed and grow very poorly. I am not asking that the authors repeat all the experiments, but the authors need to demonstrate in an example experiment that the observed effects are not influenced by the cells being stressed by the antibiotic being present in the medium.

We added a description of “when the same experiments with MMC were performed without supplementation of kanamycin, similar intensity patterns of protein bands were observed, indicating that observed effects are not influenced by the cells being stressed by the antibiotic being present in the medium” in p 2-3.

  1. The information in the strain table is insufficient. Simply stating “this study” does not tell the reader anything. The authors need to state the parent strain, the strain on which P1 was grown (for a transduction), and what was selected for. If new alleles are introduced (Datsenko and Wanner method), the authors must cross the newly-generated allele from the initial strain into a wild type background via P1 transduction, as “recombineering” procedures can result in secondary mutations. This also should be clearly reflected by the strain table.

With other modifications, we have added sentences of “The construction of mazEF::kan in BW25113 and the removal of kan from W3110N lon::kan and W3110N sulA::kan were performed with pCP20 according to the procedure described previously [37]. The first generated allele of mazEF::kan was further transferred into BW25113 by P1 transduction, generating BW25113 mazEF::kan. The construction was confirmed by PCR with the genomic DNA of the transductant as a template.” in p 10-11.

Table 2S has been revised, which includes additional several primers.

Minor points

Line 58, E. coli need to be italicised.

      corrected

Line 177, full stop missing after “sulA overexpression”.

      corrected

Other changes in the revised manuscript are corrections according to the comments of reviewer II.

Reviewer 2 Report

The manuscript by Murata et al., analyzed the function of SulA in cell cycle arrest of E.coli showing that sulA-depended cell lysis cascade pathway is different for the apoptosis-like mazEF mediated cell death pathway. In general, the provided information is satisfactory. I have some comments and suggestions:

  1. I suggest re-organize the abstract without (1) Background, (2) Methods and (3) Results, 4) Conclusions.
  2. Please explain and discuss more the selection of mitomycin C as a DNA-damaging agent.
  3. Which is the biological significance of 71 h incubation, as presented in Figure 1.
  4. From β-galactosidase activity assay is not clearly which column is related to W3110N cells harboring pBAD24 and pBAD24-SULA; I can understand that authors quantified β-galactosidase activity in vitro or not? This point should be highlighted. What are the conditions (neutral pH) and what does Z buffer contain?
  5. Can the authors explain the 32-fold lower expression level of sulA; Also the lower expression of the gene and greater protein accumulation by MMC is not well provided from Figure S2? Can the authors estimate these percentages?
  6. Figure 2 showing the morphological observation is extremely difficult to recognize the upper part. Please improve the quality and increase the size of the headers of each condition.
  7. Although the quality of PAGE gels figure are not good, please provide the exact size (kDa) of sulA (clearly show the protein band).
  8. Intracellular ROS measurement did not show any result, neither redox signaling by soxR. The authors can provide here the exact time of incubation, and the concentration of hydrogen peroxide.

Author Response

Reviewer II

Comments and Suggestions for Authors

The manuscript by Murata et al., analyzed the function of SulA in cell cycle arrest of E.coli showing that sulA-depended cell lysis cascade pathway is different for the apoptosis-like mazEFmediated cell death pathway. In general, the provided information is satisfactory. I have some comments and suggestions:

(Corrected portions in the revised manuscript are shown in blue)

  1. I suggest re-organize the abstract without (1) Background, (2) Methods and (3) Results, 4) Conclusions.

The abstract has been re-organized as suggested.

  1. Please explain and discuss more the selection of mitomycin C as a DNA-damaging agent.

Some statements related to MMC and two references [17 and 18] have been added in p 2 and p 4.

  1. Which is the biological significance of 71 h incubation, as presented in Figure 1.

We observed the recovery of CFU under MMC treatment to the level of that under the control condition. Because, the CFU level of W3110N Dlon sulA::kan did not recover to the control levels even after 48 h unlike W3110NDlon::kan as described in p 4.

  1. From β-galactosidase activity assay is not clearly which column is related to W3110N cells harboring pBAD24 and pBAD24-SULA; I can understand that authors quantified β-galactosidase activity in vitro or not? This point should be highlighted. What are the conditions (neutral pH) and what does Z buffer contain?

The open columns and gray columns have been corrected to be blue columns and red columns.

We quantified β-galactosidase activity in vitro under neutral pH. The composition of Z buffer was added.

  1. Can the authors explain the 32-fold lower expression level of sulA; Also the lower expression of the gene and greater protein accumulation by MMC is not well provided from Figure S2? Can the authors estimate these percentages?

On the basis of data that the band intensity of W3110Nlon::kan/pBAD-SULA (3 h) at 15 cycles was nearly the same as that of W3110Nlon::kan MMC (3 h) at 20 cycles, the difference of mRNA levels corresponded to difference of 5 cycles, that is, 32-fold.

Protein accumulation was shown in Figure 1, so we added Figure 1 in the sentence in p 4. Since direct comparison of protein accumulation between sulA overexpression and MMC treatment conditions is not possible, we changed “greater protein accumulation” to “similar or greater protein accumulation” in p 4.

  1. Figure 2 showing the morphological observation is extremely difficult to recognize the upper part. Please improve the quality and increase the size of the headers of each condition.

The quality and increase the size of the headers of each condition were improved.

  1. Although the quality of PAGE gels figure are not good, please provide the exact size (kDa) of sulA (clearly show the protein band).

We have added the size (kDa) of SulA in the legend of Figure 1 and an arrowhead in Figure 1.

  1. Intracellular ROS measurement did not show any result, neither redox signaling by soxR. The authors can provide here the exact time of incubation, and the concentration of hydrogen peroxide.

We added the exact time of incubation for detection of ROS. We did not measure hydrogen peroxide separately, but the prove also reacts to hydrogen peroxide (https://www.sciencedirect.com/topics/pharmacology-toxicology-and-pharmaceutical-science/dichlorodihydrofluorescein-diacetate).

Other changes in the revised manuscript are corrections according to the comments of reviewer I. New Figure S3 has been added.

Round 2

Reviewer 1 Report

This is the revised version of the study by Murata and colleagues in which the investigate how persisting SulA expression leads to cell lysis.

Overall the authors have done a good job when addressing the comments raised. There are a few more minor issues to deal with, but otherwise I believe the work is suitable for publication.

Points to address

1. The authors should go through the manuscript once more with a fine-tooth comb to look for ambiguous statements. For example, in the Abstract it is stated: “The SOS response is induced upon DNA damage and inhibition of Z-ring formation by SulA, which is encoded by an SOS response gene, allowing time for repair of damaged DNA.” As it stands this sentence implies that inhibition of Z-ring formation induces the SOS response. This is clearly not what the authors mean to say, as they state clearly and correctly that SulA is one of the LexA-regulated genes.

2. The beginning of the Results section still states some author requirements, which should be removed.

3. The authors have worked on Figure 1, panels a and b, but I fear I still find them very hard to follow. I believe changing the scale of the y axis has helped with the OD data. I am now more inclined to believe that there are some differences. However, I am still at a loss why they have squished the CFU data in the same graph. In fact, I am not sure what the CFU data are supposed to show. The axis labelling certainly is not precise. CFU in a stationary culture should be at least 2 × 109 cells per ml. Cultures should be grown from a 1:100 dilution at least, so growth should span at least two orders of magnitude. What is going on in this graph is not clear at all to me.

4. Figure 2 is much too small and should be made bigger so that details can be seen.

5. The authors have done a good job going over the nomenclature. However, once again an extra round of proof-reading is required. In line 248 both gene names should be in italics.

6. As the Discussion currently stands, especially with their model in Figure 6, the danger is that readers might imply that there is a direct link between SulA, SoxS and LpxC. The authors do not show any data confirming or disproving such a link, and I think this should be made clear by an explanatory sentence. I personally would also welcome a speculation by the authors what the link could be, but that might just be my personal taste.

7. The Data Availability Statement lacks a valid accession number.

8. The authors have tried to clarify the strain table by adding sentences in the Material & Methods section. I guess this is sufficient, but I do not find it particularly elegant, because the strain table remains of limited value. However, as the statement in the Methods section is clear I guess this will suffice.

Author Response

Comments and Suggestions for Authors

This is the revised version of the study by Murata and colleagues in which the investigate how persisting SulA expression leads to cell lysis.

Overall the authors have done a good job when addressing the comments raised. There are a few more minor issues to deal with, but otherwise I believe the work is suitable for publication.

Points to address

  1. The authors should go through the manuscript once more with a fine-tooth comb to look for ambiguous statements. For example, in the Abstract it is stated: “The SOS response is induced upon DNA damage and inhibition of Z-ring formation by SulA, which is encoded by an SOS response gene, allowing time for repair of damaged DNA.” As it stands this sentence implies that inhibition of Z-ring formation induces the SOS response. This is clearly not what the authors mean to say, as they state clearly and correctly that SulA is one of the LexA-regulated genes.

The sentence has been changed to “The SOS response is induced upon DNA damage and inhibition of Z-ring formation by SulA, whichis one of the LexA-regulated genes, allowing time for repair of damaged DNA.”

  1. The beginning of the Results section still states some author requirements, which should be removed.

The sentences have been removed.

  1. The authors have worked on Figure 1, panels a and b, but I fear I still find them very hard to follow. I believe changing the scale of the y axis has helped with the OD data. I am now more inclined to believe that there are some differences. However, I am still at a loss why they have squished the CFU data in the same graph. In fact, I am not sure what the CFU data are supposed to show. The axis labelling certainly is not precise. CFU in a stationary culture should be at least 2 × 109cells per ml. Cultures should be grown from a 1:100 dilution at least, so growth should span at least two orders of magnitude. What is going on in this graph is not clear at all to me.

Sorry, symbols (//) between 0 and 1 in the vertical axis were somehow removed and the initial OD should be 0.01.

According to the comments, the axis labelling has been re-revised and CFU data have been enlarged.

  1. Figure 2 is much too small and should be made bigger so that details can be seen.

Figure 2 has been made bigger.

  1. The authors have done a good job going over the nomenclature. However, once again an extra round of proof-reading is required. In line 248 both gene names should be in italics.

It has been done according to the indication.

  1. As the Discussion currently stands, especially with their model in Figure 6, the danger is that readers might imply that there is a direct link between SulA, SoxS and LpxC. The authors do not show any data confirming or disproving such a link, and I think this should be made clear by an explanatory sentence. I personally would also welcome a speculation by the authors what the link could be, but that might just be my personal taste.

“Further research, however, is required to show a direct link between SulA, SoxS and LpxC.” has been added in p 10.

  1. The Data Availability Statement lacks a valid accession number.

DNA microarray data were sent to Array Express. After receiving the accession number, we will inform it to the office of IJMS.

  1. The authors have tried to clarify the strain table by adding sentences in the Material & Methods section. I guess this is sufficient, but I do not find it particularly elegant, because the strain table remains of limited value. However, as the statement in the Methods section is clear I guess this will suffice.

Thank you.

Submission Date

15 March 2021

Date of this review

16 Apr 2021 19:08:58
